# Risk factors and related miRNA phenotypes of chronic pain after thoracoscopic surgery in lung adenocarcinoma patients

**Lihong Zhang<sup>☯</sup>, Liming Xu<sup>☯</sup>, Zhiyuan Chen, Haiping You, Huirong Hu, Hefan He ⓘ ***

Department of Anesthesiology, The Second Affiliated Hospital of Fujian Medical University, Quanzhou, China

☯ These authors contributed equally to this work.
* 15860905262@fjmu.edu.cn

**Data Availability Statement:** The datasets generated and/or analysed during the current study are available in the [ResMan Research Manager]

## Abstract

Chronic postsurgical pain may have a substantial impact on patient's quality of life, and has highly heterogenous presentation amongst sufferers. We aimed to explore the risk factors relating to chronic pain and the related miRNA phenotypes in patients with lung adenocarcinoma after video-assisted thoracoscopic lobectomy to identify potential biomarkers. Our prospective study involved a total of 289 patients with early invasive adenocarcinoma undergoing thoracoscopic lobotomy and a follow-up period of 3 months after surgery. Blood was collected the day before surgery for miRNA detection and patient information including operation duration, duration of continuous drainage of the chest, leukocyte count before and after operation, and postoperative pain scores were recorded. Using clinical and biochemical information for each patient, the risk factors for chronic postsurgical pain and related miRNA phenotypes were screened. We found that chronic postsurgical pain was associated with higher body mass index; greater preoperative history of chronic pain; longer postoperative drainage tube retention duration; higher numerical rating scale scores one, two, and three days after surgery; and changes in miRNA expression, namely lower expression of miRNA 146a-3p and higher expression of miRNA 550a-3p and miRNA 3613-3p in peripheral blood ($p < 0.05$). Of these factors, patient body mass index, preoperative history of chronic pain, average numerical rating scale score after operation, and preoperative peripheral blood miRNA 550a-3P expression were independent risk factors for the development of chronic postsurgical pain. Identification of individual risk markers may aid the development and selection of appropriate preventive and control measures.

## Introduction

Chronic postsurgical pain (CPSP) is defined as the pain that develops after a surgical procedure, lasts for at least 3 months, and cannot be attributed to other causes or preexisting pain [1]. Chronic pain and decline in physical function after surgery can impact patient quality of life by leading to sleep disturbances, anxiety, and depression. Thoracic surgery, including thoracoscopic surgery (although less invasive than thoracotomy), still has a high incidence of CPSP

repository, [http://www.medresman.org.cn/pub/cn/proj/projectshshow.aspx?proj=4096].

**Funding:** This work was supported by Quanzhou City Science & Technology Program of China [Grant no: 2019N105S]. And, the funders had no role in study design, data collection and analysis, decision to publish, or preparation of the manuscript.

**Competing interests:** The authors have declared that no competing interests exist.

[2, 3]. It is now generally accepted that postsurgical chronic pain is the consequence either of ongoing inflammation or a manifestation of neuropathic pain, resulting from surgical injury to major peripheral nerves [4]. Given that only a subset of surgical patients develop chronic pain, it is inferred that postoperative chronic pain presents with high individual differences and significant genetic heterogeneity [5, 6]. Existing studies have found that differentially expressed microRNAs (miRNAs) found in the peripheral blood of patients with chronic neuropathic pain are closely related and can be used as potential markers of pain [7–9].

It is helpful to use the factors associated with the development of chronic pain to formulate effective prevention and control measures. Therefore, we conducted a prospective study for over two years. The primary aim of this study was to identify independently predictors of CPSP after VATS. The second aim was to search for miRNA predictors of CPSP in peripheral blood, so as to provide directions for further research on the pathogenesis and development of CPSP.

## Methods

### Patients and inclusion criteria

This study was approved by the Ethics Committee of the Second Affiliated Hospital of Fujian Medical University (2019 Fuyi No. 2 Ethical Review [No. 208]) and has been registered in the Chinese Clinical Trial Registry, registration number: ChiCTR2200057092. Written informed consent was obtained from all subjects before study.

In the present study, 289 patients (28–79 years of age) with early invasive adenocarcinoma (diagnosed by intraoperative frozen pathology) who underwent thoracoscopic lobotomy without lymph node dissection by the same group of surgeons were observed from June 2019 to April 2022. Finally, the patients were divided into two groups (non-CPSP and CPSP) according to the presence or absence of CPSP.

The inclusion criteria were as follows: 1. American Academy of Anesthesiologists (ASA) physical status classification I–III; 2. Patients > 18 years old, with early infiltrating adenocarcinoma tissue type, with two indwelling catheters placed in the chest; 3. Absence of other malignant tumors; 4. Absence of peripheral neuropathy; 5. No peripheral (somatic) or internal (visceral) chest pain before surgery; 6. Willingness to cooperate with follow-up testing and to sign informed consent form. The exclusion criteria were as follows: 1. Transfer to open surgery for various reasons; 2. Postoperative pneumonia, atelectasis, pulmonary edema; 3. The final pathological diagnoses were different from the intraoperative frozen pathological results; 4. Those who underwent reoperation at the ipsilateral site during the study period; 5. Receiving radiotherapy, chemotherapy, or other antitumor therapy prior to or three months after surgery; 6. Study could not be completed for various reasons (failed ESP-block, loss of follow-up etc.).

### Patient characteristics and medical history

Through clinical observation and analysis of medical history, the possible factors influencing postoperative chronic pain were predicted, observed, measured, and recorded. The day before surgery, the general information and medical history for each patient were recorded, including sex, age, body mass index (BMI), ASA classification, sleep quality, anxiety state, history of chronic pain, smoking history and the presence of hypertension and diabetes. The Dosage of sufentanil, remifentanil and dexmedetomidine per kilogram during anesthesia, operation duration, postoperative drainage tube retention time, one day preoperative and one day postoperative white blood cell counts, and postoperative pain score during activity (days one, two, and three) were recorded. Peripheral blood samples were collected from patients one day

before the operation. The occurrence of CPSP was recorded at return visits via telephone three months after the operation.

## Surgery

After the patients entered the operating room and had their peripheral venous access opened, they were all given an erector spinae plane block by horizontal ultrasound of the T5 transverse process on the affected side (formula: 0.5% ropivacaine, 20 mL). Vertebral body segments were tested at 30 min after completion of the block, and if the pain of acupuncture on the blocked side was significantly less than that on the non-blocked side, then the block effect was considered satisfactory. Midazolam, cisatracurium besylate, etomidate, and sufentanil were used for general anesthesia, and a double-lumen endobronchial tube was used for tracheal intubation. Propofol, sufentanil, remifentanil, cisatracurium, and dexmedetomidine were used to maintain the general anesthesia. Oxygen saturation levels, end-tidal carbon dioxide tension, and body temperature were maintained at normal levels, while heart rate and blood pressure fluctuated to within 20% of those measurements before the operation. The bispectral index value during the operation was maintained at 40–60. After surgery, the patient was connected to an intravenous analgesia pump containing sufentanil (1 μg/(kg·d) with 0.9% saline made up to 100 mL, with a single dose of 3 mL, background dose of 1 mL/h, loading dose of 2 mL, and locking time of 15 min. The patient was awake with full Steward score and experienced no discomfort in the recovery room.

## Research tools

The Chinese validated version of the Brief Pain Inventory Short Form (BPI-SF) [10] was adopted to assess pain intensity at one, two, three, and 90 days after surgery by a trained investigator who was blinded to the patients' perioperative care to prevent any possibility of bias. After confirmation of no signs of exclusion criteria, the patients were asked whether they experienced any pain at or near the surgical area that they considered related to the thoracic surgery. Patients who replied with "no" to any of the above questions were required not to respond further questions, whereas the remaining patients were invited to finish the questionnaires. The patients were instructed to score their worst pain severity by the 0 to 10 NRS (numerical rating scale) from the preceding 24 hours. Then, the patients were inquired about the pain location and their current analgesic use. A score of 0 was considered painless, 1 to 3 indicated mild pain, 4 to 6 indicated moderate pain, and 7 to 10 indicated severe pain. According to the NRS score on the 90th day after surgery, patients with an NRS score of 0 were considered to have no CPSP, and patients with NRS score $\geq 1$ were considered to have CPSP. On days one to three after surgery, if the NRS score was $\geq 6$, 10 mg of oxycodone sustained-release tablets were given orally once every 12 h.

The State-Trait Anxiety Inventory (TAI) was used during observation. The TAI consists of 20 multiple-choice questions, with 10 questions expressing negative emotions (positive scores) and 10 questions expressing positive emotions (negative scores). Each question is scored from 1 to 4, with 1 point indicating almost never, 2 points somewhat, 3 points frequent, and 4 points almost always. According to the anxiety score, patients were divided into groups with points ranging from 20–30, 30–40, and >40. The Pittsburgh Sleep Quality Index was used to assess sleep quality. It measured seven components including subjective sleep quality, sleep latency, sleep time, sleep efficiency, sleep disorder, hypnotic drugs, and daytime dysfunction, wherein each component is scored according to grade 0 to 3, and the accumulated total score ranges from 0 to 21 points. Higher total scores indicated poorer sleep quality.

## Detection of miRNA in peripheral blood

**Collection of white blood cell samples.** One day before surgery, 5 mL of peripheral blood was collected and placed into a vacuum blood collection tube containing EDTA-K2 anticoagulant. Blood was aspirated 10 times fast through a very small needle to induce hemolysis in combination with the anticoagulant. After centrifuged,10000g, 4 degrees, 10min, the supernatant was removed to leave the white blood cell precipitate. TRIzol was added (106–107 cells plus 500ul) and aspirated repeatedly until a large amount of foam was produced. Then stored in a freezer at −80˚C.

**Extraction of total RNA.** Total RNA was extracted from white cells using mirVana$^{TM}$ RNA Isolation Kit according to the manufacturer's specifications. The white cells supplemented with TRIzol were thawed and lysed, and total RNA was extracted by chloroform extraction, isopropanol precipitation, ethanol washing, and precipitation. Firstly, added chloroform (chloroform: TRIzol = 1:5), mixed vigorously for 15s, and let stand at room temperature for 10 min. Centrifuged,12,000g, 15min, 4˚C. Absorbed the supernatant, transfered it to new EP tube, add isopropanol (isopropanol: TRIzol = 1:2), mixed thoroughly (8–10 times), and incubated at room temperature for 10min. Centrifuged,12,000g, 10min, 4˚C. It could be seen that there was gel-like precipitation at the bottom of tube. Discarded the supernatant, added 75% ethanol (ethanol: TRIzol = 1:1), and mixed gently. 7500g, 5min. Discarded the supernatant, inverted it on the filter paper and dried at room temperature for 5 min. Add 50ul RNase-free ddH20 to fully dissolve RNA. The resulting RNA solution was stored at -70˚ C. The yield of RNA was determined using a NanoDrop 2000 spectrophotometer (Thermo Scientific, USA), and the integrity was evaluated using agarose gel electrophoresis stained with ethidium bromide.

**Construction of a small RNA library and high-throughput sequencing.** Three white blood cell samples from each group treated with TRIzol were randomly selected and submitted to Shanghai Europe Easy Biomedical Technology Co., Ltd., to complete the construction of a small RNA library and high-throughput sequencing (The RNA quality was measured by 2100 Bioanalyzer and quantified using ND-2000 NanoDrop Technologies. And one case found as unqualified in quality inspection was rejected). Differences in miRNA expression between CPSP and non-CPSP groups were examined, and a total of 24 differentially expressed miRNAs were detected (p-value<0.05&|log2FC|>1) as shown in Figs 1 and 2. Of these, 6 miRNAs (miR-146a, let-7a, miR-145, miR-550a, miR-132 and miR-3613) have been reported to be associated with chronic secondary pain, especially the miR-146a [7, 11]. Combined with the analysis of data stability and difference significance, three miRNAs (two upregulated [miR-550a-3P and miR-3613-3p] and one downregulated [miR-146a-3P]) with most stable and obvious differential expression and possibly greater correlation with chronic pain were selected for real-time fluorescent quantitative polymerase chain reaction (PCR) verification.

**Fluorescent quantitative PCR detection.** Quantification was performed with a two-step reaction process: reverse transcription (RT) and PCR. A miRNA reverse transcription kit (No: AT351, TransScript miRNA First-Strand cDNA Synthesis SuperMIX, TransGen Biotech, China) was used to conduct reverse transcription reaction on the extracted total RNA, following the manufacturers protocol. Each RT reaction consisted of 0.5 μg RNA, 5 μl of 2×TS miRNA Reaction Mix and 0.5 μl of *TransScrip* miRNA RT Enzyme Mix, in a total volume of 10 μl. Reactions were performed in a GeneAmp$^®$ PCR System 9700 (Applied Biosystems, USA) for 60 min at 37˚C, followed by heat inactivation of RT for 5s at 85˚C. The 10 μl RT reaction mix was then diluted × 10 in nuclease-free water. Real-time PCR was performed using LightCycler$^®$ 480 Ⅱ Real-time PCR Instrument (Roche, Swiss) with 10 μl PCR reaction

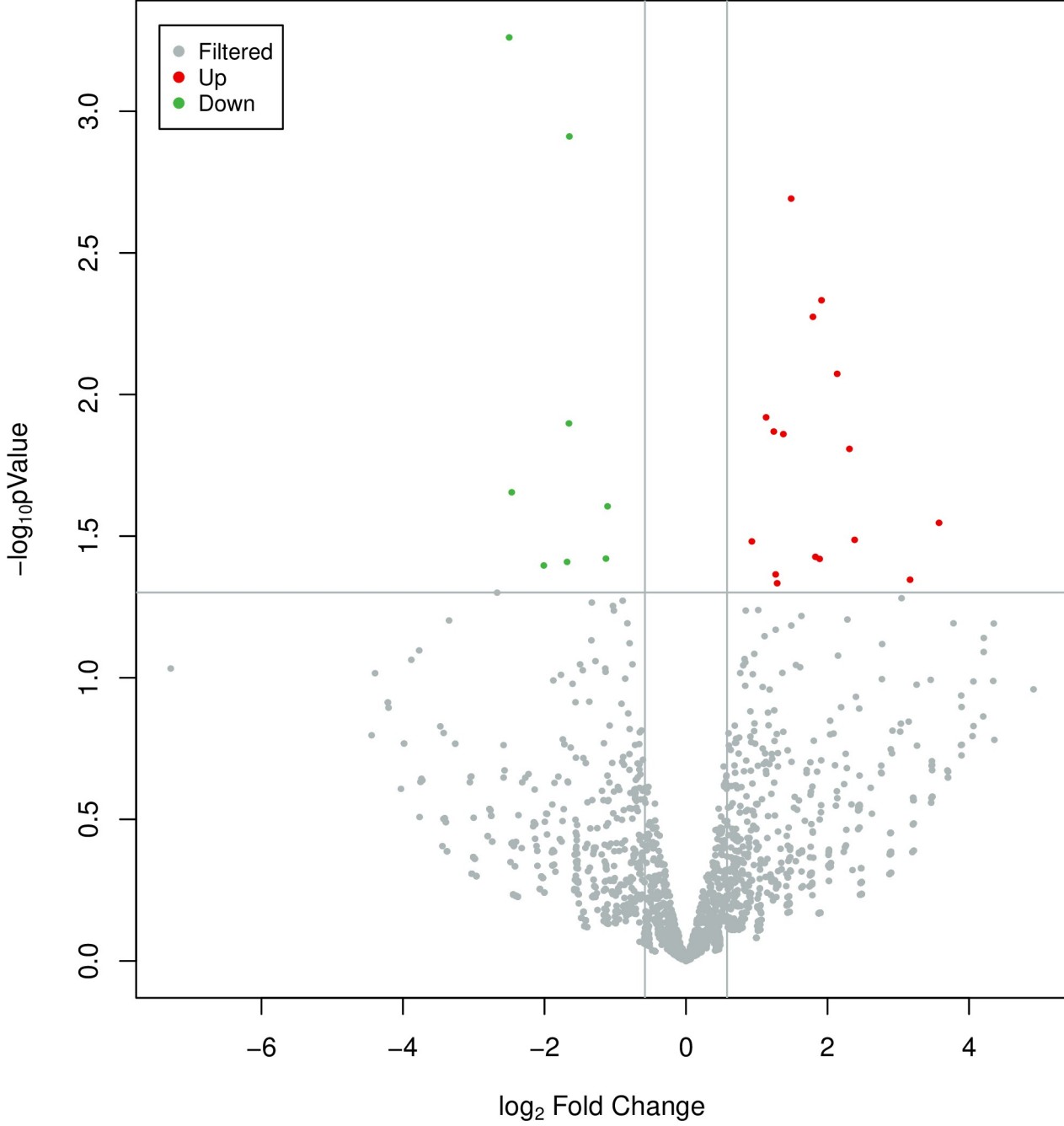

**Fig 1. CPSP-vs-Control-volcano-pval-0.05-FC-2.** miRNA.

mixture that included 1 μl of cDNA, 5 μl of 2×PerfectStartTM Green qPCR SuperMix, 0.2μl of universal primer, 0.2 μl of microRNA-specific primer and 3.6 μl of nuclease-free water. Reactions were incubated in a 384-well optical plate (Roche, Swiss) at 94˚C for 30s, followed by 45 cycles of 94˚C for 5s, 60˚C for 30s. Each sample was run in triplicate for analysis. At the end of

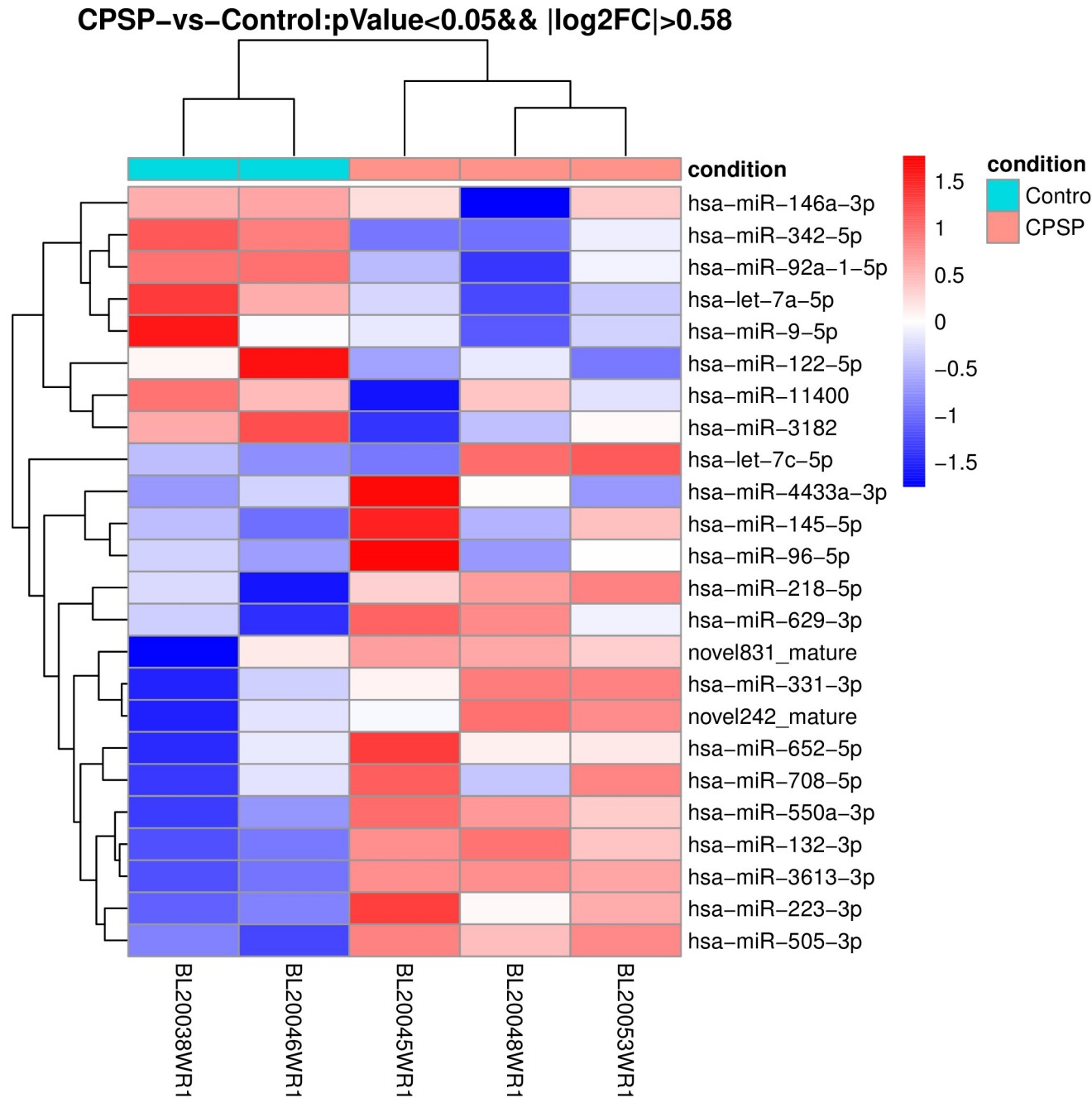

**Fig 2. CPSP-vs-Control-heatmap-pval-0.05-FC-2.** miRNA.

the PCR cycles, melting curve analysis was performed to validate the specific generation of the expected PCR product. *U6* was selected as the internal reference gene. The primer sequences are listed in Table 1. The relative expression levels were determined using the $2^{-\triangle\triangle Ct}$ method. Finally, Blood samples from 232 patients (88 in the CPSP group and 144 in the non-CPSP group) were used for miRNA detection and the Ct values of all the samples are provided in S1 Table.

**Table 1. Types of miRNAs.**

| miRNA | Forward primer (5′→3′) |
|---|---|
| U6 | CAAGGATGACACGCAAATTCG |
| miR-550a-3P | TCTTACTCCCTCAGGCACAT |
| miR-146a-3P | TCTGAAATTCAGTTCTTCAGAAA |
| miR-3613-3P | CGACAAAAAAAAAGCCCAACC |

## Sample size

The sample size was calculated using the minimal clinically difference found in our previous study (History of Chronic Pain in CPSP group: 0.2885 and non-CPSP group:0.1294) [12]. And yielded the following results: 95 cases in the CPSP group and 155 cases in the non-CPSP group ($\alpha = 0.05$; power = 0.9; ratio of sample size: 0.61; two-sided; lost to follow-up ratio: 0.1). Additionally, there should be an adequate number of events per independent variable to avoid an overfit model, with commonly recommended minimum "rules of thumb" ranging from 10 to 20 events per covariate [13]. Therefore, 289 patients were observed (Seven covariates were included in the regression analysis). Due to loss of follow-up and other reasons, 57 cases were excluded. Finally, 232 patients were included (88 patients in the CPSP group and 144 patients in the non-CPSP group).

## Statistical analysis

The normality of continuous data was tested using the Shapiro–Wilk test. Normally distributed parameters are presented as ($\bar{x} \pm s$) and were analyzed using the Student's t-test. Non-normally distributed parameters are presented as median M (P25, P75) and were analyzed using the Mann–Whitney U test. The difference in continuous variables over time was tested using two-way repeated measures ANOVA. Categorical data were described as numbers or percentages and analyzed with the $\chi 2$ test. Risk factor analysis of CPSP was performed using binary logistic regression.

Statistical significance was defined as $p < 0.05$. SPSS Statistics version 26.0 for Windows was used to perform all analyses.

## Results

At last, 57 patients were excluded by application of the exclusion criteria, 8 of them were transferred to open surgery, 16 of them developed postoperative pneumonia, atelectasis, pulmonary edema, 5 of them were founded to be different final pathological diagnoses from intraoperative frozen, and 28 of them were excluded because they were unable to complete the study for various reasons. Finally, a total of 232 eligible patients were included in the study (Fig 3). Of these, 88 experienced CPSP (37.9%) and 144 did not. CPSP was mild in 79 cases (89.8%), moderate in eight cases (9%), and severe in one case (1.2%, with affected sleep).

As shown in Table 2, the CPSP group had a higher BMI, greater history of preoperative chronic pain, and longer duration of continuous tube drainage than the non-CPSP group ($p < 0.05$). There were no significant differences in sex, age, sleep quality score, trait anxiety score, history of hypertension, history of diabetes, operation time, or WBC count difference before and after surgery between the two groups (P > 0.05).

There was an intergroup difference in the NRS scores between the two groups after surgery ($p = 0.001$), and the NRS scores in the CPSP group were higher than those in the non-CPSP groups. The differences in NRS scores at the three time points were statistically significant

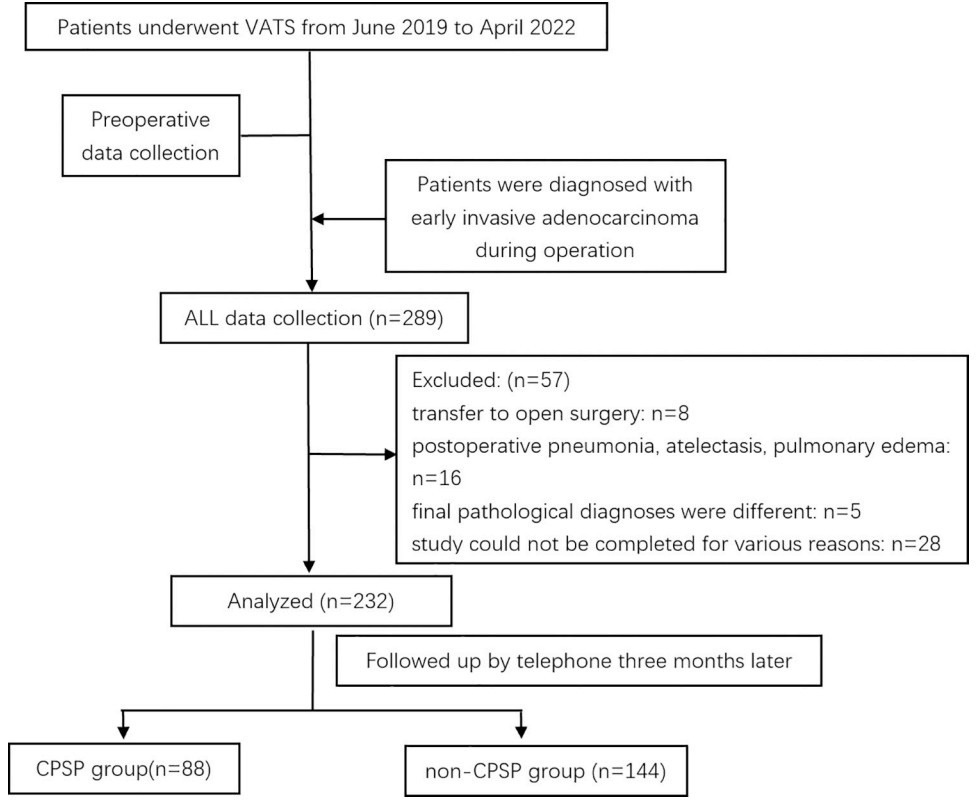

**Fig 3. Flow diagram for patient inclusion.**

**Table 2. Comparison of general and clinical information between the two groups.**

| General information | CPSP[†] group (n = 88) | Non-CPSP[†] group (n = 144) | t/Z/χ² | p |
|---|---|---|---|---|
| Male/female | 38/50 | 71/73 | 0.822 | 0.365 |
| Age (years) | 55.41 ± 10.09 | 57.04 ± 10.76 | -1.147 | 0.252 |
| Body mass index | 24.28 ± 2.90 | 22.79 ± 2.45 | 4.160 | 0.000 |
| Sleep quality score (score) | 5.52 ± 3.91 | 6.21 ± 3.38 | -1.413 | 0.159 |
| Trait anxiety score (score) | 29.77 ± 6.16 | 30.01 ± 6.55 | -0.278 | 0.781 |
| Hypertension (with/without) | 22/66 | 33/111 | 0.131 | 0.717 |
| Diabetes (with/without) | 9/79 | 14/130 | 0.016 | 0.901 |
| History of chronic pain, n (%) | 25 (28.41) | 18 (12.50) | 9.156 | 0.002 |
| Smoking history, n (%) | 11(12.50) | 27(18.75) | 1.558 | 0.212 |
| Sufentanil dosage (ug/kg) | 0.83 (0.75, 0.83) | 0.83 (0.67, 0.83) | 1.160 | 0.246 |
| Remifentanil dosage (ug/kg) | 16 (12,20) | 15(11,20) | 1.526 | 0.127 |
| Dexmedetomidine dosage (ug/kg) | 0.9 (0.7,1.0) | 0.8 (0.7,1.0) | 1.340 | 0.180 |
| Operation duration (min) | 204.20 ± 74.30 | 187.64 ± 70.52 | 1.700 | 0.090 |
| Continuous drainage time of the chest (days) | 3.71 ± 1.38 | 3.31 ± 0.69 | 2.551 | 0.012 |
| White blood cell count difference (×10⁹/L) | 6.31 ± 2.36 | 5.68 ± 3.12 | 1.622 | 0.106 |

[†]CPSP, chronic postsurgical pain

**Table 3. Comparison of numerical rating scale scores after operation between the two groups, ($\bar{x}$±s, score).**

| Group | n | NRS[‡] 1 d | NRS[‡] 2 d | NRS[‡] 3 d | NRS[‡] average |
|---|---|---|---|---|---|
| CPSP[†] group | 88 | 3.57 ± 1.37 | 2.81 ± 1.16 | 2.12 ± 0.96 | 2.34±0.73 |
| Non-CPSP group | 144 | 2.94 ± 1.06 | 2.35 ± 0.93 | 1.74 ± 0.71 | 2.83±0.99 |
| | | Time comparison | Group comparison | Interaction group* time | |
| F/t | - | 183.329 | 18.884 | 1.646 | 4.04 |
| p | - | 0.000 | 0.000 | 0.197 | 0.000 |

[†]CPSP, chronic postsurgical pain

[‡]NRS, numerical rating scale.

($p$ = 0.001). The longer the operation time, the lower the NRS score (Table 3). However, there was no significant differences in the interaction effects between groups and time.

Compared with the non-CPSP group, there were 16 upregulated and eight downregulated miRNAs in the CPSP group, as shown in Figs 1 and 2. Compared to that of the non-CPSP group, the preoperative expression of miR-146a-3P in the peripheral blood of the CPSP group was lower, and the expression of miR-550a-3P and miR-3613-3P was higher ($p < 0.05$), as shown in Table 4.

Binary logistic regression was applied to examine predictors of CPSP after VATS. The presence of CPSP was regarded as the dependent variable and covariates were chosen based on statistical significance or possible clinical importance, variates with P<0.10 in the univariate analysis were entered in the regression analysis. In addition, collinearity diagnosis on all the covariables included in the regression analysis was performed and found that the tolerance>0.2 and VIF<5. It could be considered that there was no collinearity as shown in Table 5. As showed in Table 6, four risk factors were identified for CPSP after VATS: BMI (OR 1.207, 95% CI 1.069–1.363, P = 0.002), preoperative history of chronic pain (OR 2.865, 95% CI 1.321–6.215, P = 0.008), average NRS score (OR1.838, 95% CI 1.266–2.668, P = 0.001), and preoperative peripheral blood miR-550a-3P value (OR1.699, 95% CI 1.242–2.324, P = 0.001) The correlation between observed factors and the occurrence of CPSP is shown in Fig 4. The prediction probability for CPSP after VATS yield the area under the receiver operating characteristic curve of 0.781 (95% CI 0.718–0.844) (Fig 5), and the model showed good calibration by Hosmer–Lemeshow goodness-of-fit statistic with $\chi^2$ = 4.317, P = 0.827.

## Discussion

The incidence, influencing factors, and pathogenesis of CPSP may vary significantly among different patients undergoing different types of surgeries.

In this study, preoperative erector spinae plane block was performed to reduce the degree of acute postoperative pain. The difference of white blood cells before and after the operation did

**Table 4. Comparison of miRNA in preoperative peripheral blood of patients, M (P25, P75).**

| Group | n | miR 146a-3P | miR 550a-3P | miR 3613-3P |
|---|---|---|---|---|
| CPSP[†] group | 88 | 0.64 (0.25, 1.81) | 1.42 (0.79, 2.60) | 1.18 (0.82, 1.72) |
| Non-CPSP[†] group | 144 | 1.02 (0.46, 2.28) | 1.03 (0.59, 1.74) | 0.93 (0.65, 1.47) |
| Z | - | 2.214 | 3.303 | 2.336 |
| p | - | 0.027 | 0.001 | 0.020 |

[†]CPSP, chronic postsurgical pain.

**Table 5. Collinearity diagnosis.**

| model | | coefficient<sup>a</sup> | | | | | | | |
|---|---|---|---|---|---|---|---|---|---|
| | | Unstandardized coefficient | | Standardized coefficient | - | - | Collinearity statistics | |
| | | B | Standard error | Beta | t | Significance | Tolerance | VIF |
| 1 | (constant) | -1.175 | 0.275 | - | -4.277 | 0.000 | - | - |
| | BMI | 0.034 | 0.011 | 0.189 | 3.116 | 0.002 | 0.937 | 1.067 |
| | Continuous drainage time | 0.049 | 0.029 | 0.105 | 1.699 | 0.091 | 0.912 | 1.097 |
| | History of chronic pain | 0.196 | 0.074 | 0.157 | 2.635 | 0.009 | 0.971 | 1.030 |
| | Operation duration | 0.000 | 0.000 | 0.072 | 1.184 | 0.237 | 0.935 | 1.070 |
| | NRS[†] average | 0.115 | 0.034 | 0.205 | 3.399 | 0.001 | 0.949 | 1.054 |
| | miR 146a-3P | -0.009 | 0.009 | -0.065 | -1.096 | 0.274 | 0.979 | 1.021 |
| | miR 550a-3P | 0.093 | 0.027 | 0.206 | 3.456 | 0.001 | 0.972 | 1.028 |
| | miR 3613-3P | 0.045 | 0.040 | 0.067 | 1.121 | 0.264 | 0.969 | 1.032 |

a. Dependent variable: group

not have statistical significance could exclude the effect of postoperative infection on CPSP. All patients with early lung adenocarcinoma as their pathological diagnosis underwent video-assisted thoracoscopic lobectomy without lymph node dissection by the same group of surgeons. And the results also showed no significant difference on surgery duration. The control of these perioperative factors that may affect the occurrence of CPSP was conducive to search for CPSP susceptible population and find more closely related and more predictive genetic factors.

Meanwhile, considering the influence of some possible changing factors on the development of CPSP during the study (It is now generally accepted that postsurgical chronic pain is the consequence either of ongoing inflammation or a manifestation of neuropathic pain, resulting from surgical injury to major peripheral nerves [4]. In addition, a 10-year single-center retrospective study showed that postoperative chemotherapy, and postoperative radiotherapy were significant risk factors for CPSP [3]),and the consistency and integrity of the collected data. 57 patients (= 19.7%) were excluded by application of the exclusion criteria.

MicroRNAs (miRNAs) are small, endogenous, non-coding RNAs of $\sim$ 22 to 26 nucleotides in length that play important regulatory roles in a substantial proportion of processes in both normal and disease states [14]. Some studies have shown that differential expression of miRNAs is closely associated with chronic pain [7–9]. Microarray and deep-sequencing analyses revealed that nerve injury or noxious stimuli could induce broad changes in miRNA expression in serum or along the pain processing pathways. Dysregulated miRNAs contribute to neuropathic pain via

**Table 6. Logistic regression analysis of risk factors for chronic postsurgical pain.**

| Single factor | β | Standard error | Wald $\chi^2$ | OR[‡] (95% CI) | p |
|---|---|---|---|---|---|
| Body mass index | 0.188 | 0.062 | 9.230 | 1.207 (1.069, 1.363) | 0.002 |
| Continuous drainage time | 0.289 | 0.178 | 2.650 | 1.335 (0.943, 1.891) | 0.104 |
| History of chronic pain | 1.053 | 0.395 | 7.097 | 2.865 (1.321,6.215) | 0.008 |
| Operation duration | 0.003 | 0.002 | 1.498 | 1.003 (.998,1.007) | 0.221 |
| NRS[†] average | 0.609 | 0.190 | 10.239 | 1.838 (1.266, 2.668) | 0.001 |
| miR 146a-3P | -0.049 | 0.049 | 1.005 | 0.952 (0.866, 1.048) | 0.316 |
| miR 550a-3P | 0.530 | 0.160 | 10.989 | 1.699(1.242, 2.324) | 0.001 |
| miR 3613-3P | 0.241 | 0.212 | 1.288 | 1.272(0.840, 1.927) | 0.256 |

NRS[†], numerical rating scale

OR[‡], odds Ratio.

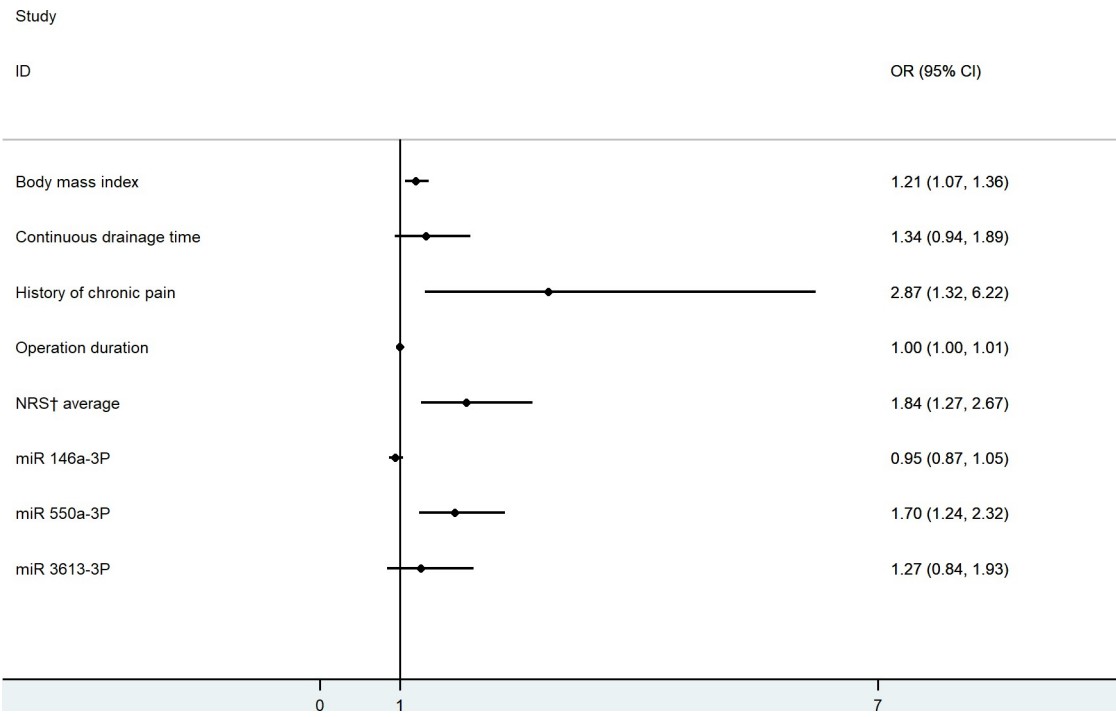

**Fig 4. CPSP-vs-Control-forest-map.**

neuroinflammation, autophagy, abnormal ion channel expression, regulating pain-related mediators, protein kinases, structural proteins, neurotransmission excitatory–inhibitory imbalances, or exosome miRNA-mediated neuron–glia communication [15]. There are several studies reporting changes in miRNA expression in patients with chronic pain, such as miR-146a, let-7a, miR-145, miR-550a, miR-132 and miR-3613 [7, 11]. Therefore, in this study, we analyzed the differential expression of the miRNAs 146a-3P, 550a-3P, and 3613-3P, which were initially identified as relevant to CPSP through rough micro-RNA sequencing.

In this prospective study, the incidence of CPSP three months after surgery was 37.9%. Logistic regression analysis showed that high BMI, preoperative history of chronic pain, average NRS score after operation, and high expression of miR-550a-3P in preoperative peripheral blood were the risk factors for the development of CPSP. It has been reported that obese patients after surgical treatment of lung cancer suffer more from pain in the postoperative period than nonobese patients [16]. In another study, obesity was found to be linked with genetic polymorphism altering sensitivity to pain, thus indicating that chronic pain might be related both to obesity and genetic factors [17]. Resistin is considered a proinflammatory cytokine that is primarily expressed and secreted by monocytes and macrophages in humans [18]. High BMI as a risk factor for developing CPSP may be related to the participation of resistin in adipose tissue in the regulation of inflammatory processes [19].

There are differences in opinions in the literature on whether the postoperative pain score is a risk factor for developing CPSP [20–22]. In the present study, the NRS scores of patients in the two groups were significantly different on one, two, and three days after surgery, and the acute postoperative pain intensity of patients in the CPSP group was higher than that of patients without CPSP. In addition, according to regression analysis results, the mean NRS score 3 days after operation was a risk factor for CPSP. Consequently, clinicians should take active measures to control postoperative acute pain in patients, which could facilitate the prevention of the occurrence

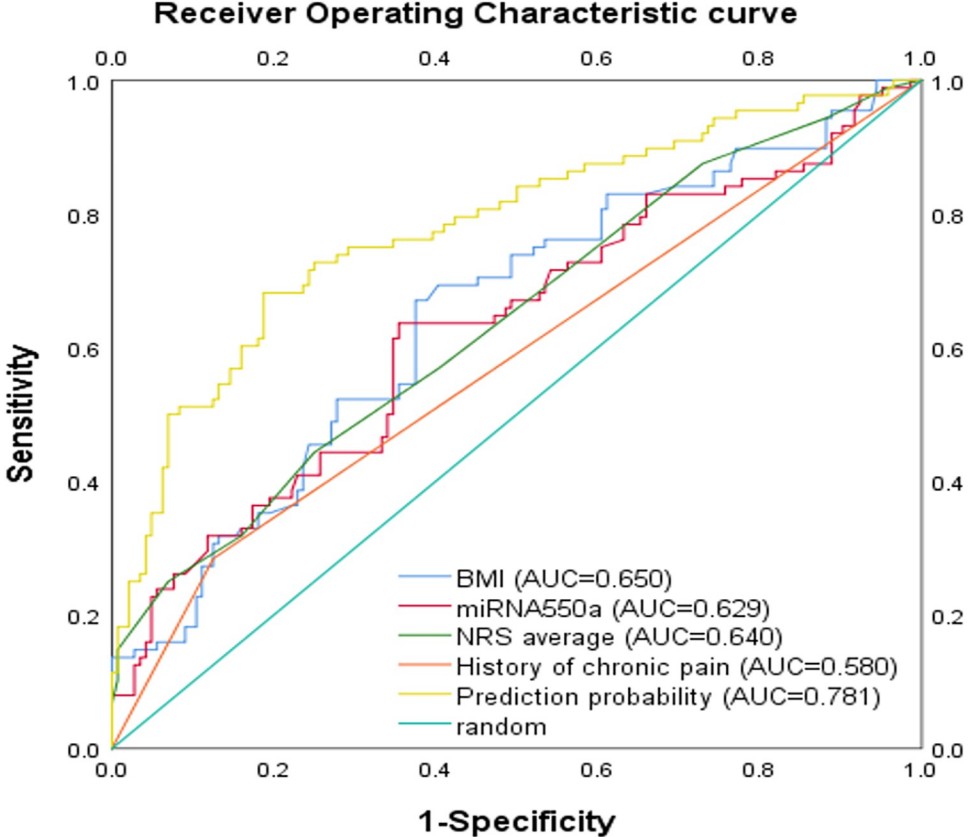

**Fig 5. The area under the ROC curve of CPSP model.**

of postoperative chronic pain. The study showed the early warning effect of miR-550a-3P on CPSP after thoracoscopic surgery. Pellegrino A et al. also showed strong correlation between miR-550a-3p and chronic painful polyneuropathies [11]. Their research demonstrated that gene "myelin transcription factor 1 like" (MYT1L) is potentially targeted by miR-550a-3p [11]. MYT1L promotes axonal development/differentiation, neurite outgrowth/proliferation, synaptic transmission, extracellular matrix composition, as well as remyelination after induced demyelination [23, 24]. In this study, the expression levels of miR-146a-3P and miR-3613-3P in preoperative peripheral blood of patients without CPSP were significantly different from those with CPSP. Phạm TL et al. demonstrated that miR-146a-5p-loaded nanoparticles (NPs) can attenuate neuropathic pain behaviors in the rat spinal nerve ligation-induced neuropathic pain model by inhibiting activation of the NF-κB and p38 MAPK pathways in spinal microglia [25]. And certain study has shown that miR-3613-3P is highly expressed in the X chromosome and is related to sex-related pain [26], which may explain why relevant research shows that women are at a higher risk for developing CPSP [21]. But the regression analysis did not support that miR-146a-3P and miR-3613-3P were the risk factors for developing CPSP in these patients. The specific reasons for the differences from the previous reports need to be further studied.

The present study had some limitations. First, it is difficult to guarantee the accuracy of a follow-up telephone interview to determine whether patients have CPSP and the severity of CPSP. Second, the observations were limited to patients undergoing thoracoscopic surgery for early lung adenocarcinoma, meaning it is not clear whether the results can be generalized to other lung cancer patients. Third, no pathway analysis associated with microRNAs. To obtain

more accurate and applicable conclusions, further studies should be carried out which expand the number of observational cases and improve the observation methods.

In summary, the present study reveals that the preoperative BMI, preoperative expression level of miR-550a-3P in peripheral blood, preoperative history of chronic pain, and postoperative NRS score are independent risk factors for CPSP following thoracoscopic lobotomy in patients with early-stage pulmonary infiltrating adenocarcinoma. In the case of patients with high preoperative BMI, high expression of miR-550a-3P in peripheral blood, or a history of chronic pain, clinicians should actively take measures to reduce postoperative NRS scores, which could reduce the incidence of postoperative chronic pain. If an obese patient has long-lasting chronic pain after surgery, can it be treated by losing weight? And the expression difference of miR-550a-3P may be one of the reasons for the genetic susceptibility to CPSP in patients with thoracoscopic lobectomy, which needs to be confirmed by further studies in the future. More in-depth studies on potential mechanisms of CPSP development are needed for the prevention and treatment of CPSP in these patients.

## Supporting information

**S1 Table. Ct values of miR-146a-3P, miR-550a-3P and miR-3613-3P in CPSP group and non-CPSP group.**
(TIF)

## Author Contributions

**Conceptualization:** Lihong Zhang, Hefan He.

**Data curation:** Liming Xu.

**Formal analysis:** Huirong Hu.

**Methodology:** Haiping You.

**Resources:** Hefan He.

**Writing – original draft:** Lihong Zhang, Zhiyuan Chen, Hefan He.

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
