## [Decision Letter · Decision Letter 0]

28 Jun 2023

PONE-D-23-13874Risk factors and related miRNA phenotypes of chronic pain after thoracoscopic surgery in lung adenocarcinoma patientsPLOS ONE

Dear Dr. He,

Thank you for submitting your manuscript to PLOS ONE. After careful consideration, we feel that it has merit but does not fully meet PLOS ONE’s publication criteria as it currently stands. Therefore, we invite you to submit a revised version of the manuscript that addresses the points raised during the review process.

ACADEMIC EDITOR:

Please revise the manuscript according to the reviewers suggestions, 

We look forward to receiving your revised manuscript.

Kind regards,

Silvia Fiorelli

Academic Editor

PLOS ONE

Reviewers' comments:

Reviewer's Responses to Questions

**Comments to the Author**

1. Is the manuscript technically sound, and do the data support the conclusions?

Reviewer #1: Yes

Reviewer #2: Partly

2. Has the statistical analysis been performed appropriately and rigorously? 

Reviewer #1: No

Reviewer #2: No

3. Have the authors made all data underlying the findings in their manuscript fully available?

Reviewer #1: Yes

Reviewer #2: Yes

4. Is the manuscript presented in an intelligible fashion and written in standard English?

Reviewer #1: Yes

Reviewer #2: Yes

5. Review Comments to the Author

Reviewer #1: The paper by He and colleagues describes the analysis of patients with adenocacinoma. Specifically, the miRNA of the patients is analysed to identify potential biomarkers.

Before the paper can be accepted, the following points should be addressed:

- The number of citations and the selected citations do not adequately reflect the current state of research. The authors should consider current literature

- A description of how pain was measured and classified is completely missing. There are several accepted standards in research. Which of these was used and what the results. This data are needed to be added. Furthermore, a correlation of these parameters with the miRNA profile should be done.

- The description of the methods is insufficient. E.g. page 7 line 123-124 "After an appropriate amount of TRIZOL ...". This is not a scientific description that allows repeating the experiments. For each experimental step, a detailed protocol should be given as stated in the Authorguidelines.

- Page 7: Construction of miRNA Library

o Number of samples is missing

o Quality control of each step is missing

o Why was the miRNA not isolated from serum?

o Which threshold for differentially expressed miRNA was chosen?

o How was "possibly greater correlation with chronic pain" determined?

- Page 8: qPCR

o Why was total RNA used for qPCR? There is a risk that the primers are not specific.

o The sequence of the reverse primer used is missing

o qPCR conditions are missing and a table containing the Ct values of the samples

- Page 8: statistical analysis

o How was the necessary sample size (number of patients) calculated?

o Information on which patients were excluded / included

Results:

- The distribution of patients between the two groups is uneven e.g. the CPSP group contains more women.

- How was it determined that differences in clinical information did not affect the data? What mathematical models were used?

- Were the data analysed by further subdividing the groups e.g. male samples only, young patients vs older patients, etc.?

Discussion:

- On page 13, line 240 it is stated that after surgery 37.9% of patients developed CPSP. What is the normal range?

- The discussion lacks any description of the biochemical function and pathways involved of the miRNAs. At least one GO annotation and appropriate discussion needs to be made

- Which miRNAs have been identified / described in patients with early adenoma carcinoma?

- Which miRNAs were identified / described in patients with chronic pain?

- The miRNA part of the discussion is a collection of known facts of the identified miRNAs but no discussion. The whole section needs to be revised.

Reviewer #2: The authors need to make it very clear that correlation does not mean causation - this is not clear.

Second, authors need to clarify whether the factors identified are correlated to each other or how the statistical/correlation analyses were performed and corrected for other co-variables.

6. PLOS authors have the option to publish the peer review history of their article (what does this mean?). If published, this will include your full peer review and any attached files.

Reviewer #1: No

Reviewer #2: No

---

## [Author Response · Author response to Decision Letter 0]

20 Aug 2023

General response: We thank Editor and Reviewers for both the time and the insightful comments. We have carefully taken into consideration all the comments, and revised the manuscript substantially by adding analysis and clarifying content to make it more scientific and innovative. We have addressed all questions in a point-by-point manner, as shown below.

Response: Thanks for your reminder and the handy link. We have read PLOS ONE Formatting guidelines carefully and made sure that our manuscript meets PLOS ONE's style requirements as the manuscript shows.

Response: We are very sorry for our negligence of providing the wrong the message on the 'Financial Disclosure' section. We have corrected the Financial Disclosure to make the grant information consistent in the 'Funding Information' and 'Financial Disclosure'. And we have updated the statement in our cover letter.

Reviewers’ comments

Reviewer#1

Comment No.1: The number of citations and the selected citations do not adequately reflect the current state of research. The authors should consider current literature. Response: We agree wholeheartedly that our manuscript needs to add more current literature. In the revised manuscript, we have updated the literature by replacing the older citations and adding some more recent literature to our analysis, such as Sabina S et al. (2022), Jin J et al. (2022), Chen WC et al. (2022), Jiang M et al. (2022), Zhang Y et al. (2022), Pellegrino A et al. (2023), Chen J et al. (2022) and Shi, Y et al. (2018), Phạm TL et al. (2020).

Comment No.2: A description of how pain was measured and classified is completely missing. There are several accepted standards in research. Which of these was used and what the results. This data is needed to be added.

Response: We are very grateful to the Reviewers for their constructive suggestions. We only studied whether patients had CPSP after surgery, Thus, the patients were inquired only about the worst pain intensity, pain location and their current analgesic use. We have added Chinese validated version of the BPI-SF (Brief Pain Inventory Short Form) to assess pain intensity. And the description of how pain was measured and classified has been completed. (Page 6, lines 101–116)

Comment No.3: Furthermore, a correlation of these parameters with the miRNA profile should be done.

Response: We fully agree that a correlation analysis of the parameters should be done. We performed collinearity diagnosis on all the covariables included in the regression analysis and found that the tolerance＞0.2 and VIF＜5. It could be considered that there was no collinearity as shown in Table 5. (page 14, Table 5)

Comment No.4: The description of the methods is insufficient. E.g., page 7 line 123-124 "After an appropriate amount of TRIZOL ...". This is not a scientific description that allows repeating the experiments. For each experimental step, a detailed protocol should be given as stated in the Author guidelines.

Response: We are very sorry for our negligence of our experimental procedure description. We have already added the description for each experimental step. (page7-10, lines 130–188)

Comment No.5: Page 7: Construction of miRNA Library

1. Number of samples is missing

2. Quality control of each step is missing

3. Why was the miRNA not isolated from serum?

4. Which threshold for differentially expressed miRNA was chosen?

5. How was "possibly greater correlation with chronic pain" determined?

Response: （For questions 1 and 5）We are sorry that we may have not expressed clearly about the number of sample and the reason for the chosen miRNAs which were possibly greater correlative with chronic pain. 

Actually, three white blood cell samples from each group treated with TRIzol were randomly selected and submitted to Shanghai Europe Easy Biomedical Technology Co., Ltd., to complete the construction of a small RNA library and high-throughput sequencing. (And one case found as unqualified in quality inspection was rejected.) (page8, lines 153–157)

The steps on how to select miRNAs which are more correlative with chronic pain are as follows: Firstly, we completed the construction of a small RNA library and high-throughput sequencing. Secondly, we detected a total of 24 differentially expressed miRNAs (p-value<0.05&|log2FC|>1). Thirdly, we selected three miRNAs with obvious differential expression through atlas analysis and literature search [6,8]. And the above expression has been placed in the ‘Construction of a small RNA library and high-throughput sequencing’ section. (page8, lines 153–164)

（For question 2）We fully agree that the quality control for each step should be supplemented. We have added quality control to the extracted total RNA (page8, lines 148–150,155-157) and the PCR detection (page9, lines 183–185).

（For question 3）It is really true as Reviewer mentioned that the miRNA we got was not from serum. Because the miRNA in serum may come from both intracellular and extracellular sources, such as red and white blood cells. The amount of miRNA directly extracted from cells will be more. Moreover, one of the mechanisms leading to chronic pain is inflammation, and white blood cells are closely related to inflammation, so we chose white blood cells to extract miRNA to study the expression difference between the two groups.

（For question 4）We are grateful to the Reviewer for the reminder. Actually, the threshold for differentially expressed miRNA was p-value<0.05 and |log2FC|>1. (page8, line 159).

Comment No.6: Page 8: qPCR

o Why was total RNA used for qPCR? There is a risk that the primers are not specific.

o The sequence of the reverse primer used is missing

o qPCR conditions are missing and a table containing the Ct values of the samples

Response: We very much appreciate the Reviewer’s valuable comments and suggestions. The design and synthesis of primer sequences for the three miRNAs were specific which can be found in Table 1 of the text. And the reverse primers came with kits, but we are sorry that the reverse sequences couldn’t be provided due to confidentiality of the other party. We have added complete qPCR conditions. (page9-10, lines 169–188) And the Ct values of all the samples have been provided in S1 Table. 

Comment No.7: statistical analysis

o How was the necessary sample size (number of patients) calculated?

o Information on which patients were excluded / included

Response: We are very grateful to the Reviewer for the constructive suggestions. We have added detailed basis for sample size calculation, and list them separately in the "Sample size" section. (page10, lines 191–201) And we have placed the included / excluded criteria in the’ Patients and inclusion criteria’ section. (page4, lines 60–69)

Comment No.8: Results:

- The distribution of patients between the two groups is uneven e.g., the CPSP group contains more women.

- How was it determined that differences in clinical information did not affect the data? What mathematical models were used?

- Were the data analysed by further subdividing the groups e.g., male samples only, young patients vs older patients, etc.?

Response: Thanks for Reviewer’s questions. We aimed to explore the risk factors relating to chronic postsurgical pain (CPSP). And the patients were divided into two groups (non-CPSP and CPSP) according to the presence or absence of CPSP. Therefore, there existed some differences in clinical information which were the variables we were looking for. It is really true that the CPSP group contains more women, but there are no significant differences in sex between the two groups. If there was a significant difference, then we would include it in the regression analysis to determine the risk factor.

Comment No.9: Discussion:

- On page 13, line 240 it is stated that after surgery 37.9% of patients developed CPSP. What is the normal range?

- The discussion lacks any description of the biochemical function and pathways involved of the miRNAs. At least one GO annotation and appropriate discussion needs to be made

- Which miRNAs have been identified / described in patients with early adenoma carcinoma?

- Which miRNAs were identified / described in patients with chronic pain?

- The miRNA part of the discussion is a collection of known facts of the identified miRNAs but no discussion. The whole section needs to be revised.

Response: We very much appreciate the Reviewer’s valuable comments and suggestions. According to the literature, the incidence of CPSP after thoracic surgery is 25-75%. (Chen WC, Bai YY, Zhang LH, et al. Prevalence and Predictors of Chronic Postsurgical Pain After Video-Assisted Thoracoscopic Surgery: A Systematic Review and Meta-analysis. Pain , 2023,12(1):117-139. doi: 10.1007/s40122-022-00439-0.）And we have re-written this part according to the Reviewer’s suggestion. (page15,16,17)

Reviewer#2

Comment: The authors need to make it very clear that correlation does not mean causation - this is not clear.

Second, authors need to clarify whether the factors identified are correlated to each other or how the statistical/correlation analyses were performed and corrected for other co-variables.

Response: We very much appreciate the Reviewer’s valuable comments and suggestions. We fully agree that correlation does not mean causation. We have performed collinearity diagnosis on all the covariables included in the regression analysis and found that the tolerance＞0.2 and VIF＜5. It could be considered that there was no collinearity as shown in Table 5. (page 14, Table 5)

---

## [Decision Letter · Decision Letter 1]

4 Oct 2023

PONE-D-23-13874R1Risk factors and related miRNA phenotypes of chronic pain after thoracoscopic surgery in lung adenocarcinoma patientsPLOS ONE

Dear Dr. He,

Thank you for submitting your manuscript to PLOS ONE. After careful consideration, we feel that it has merit but does not fully meet PLOS ONE’s publication criteria as it currently stands. Therefore, we invite you to submit a revised version of the manuscript that addresses the points raised during the review process.

ACADEMIC EDITOR:please carefully assess all the reviewers comments

Please submit your revised manuscript by Nov 18 2023 11:59PM  If you will need more time than this to complete your revisions, please reply to this message or contact the journal office at plosone@plos.org. Please include the following items when submitting your revised manuscript:A rebuttal letter that responds to each point raised by the academic editor and reviewer(s). You should upload this letter as a separate file labeled 'Response to Reviewers'.A marked-up copy of your manuscript that highlights changes made to the original version. You should upload this as a separate file labeled 'Revised Manuscript with Track Changes'.An unmarked version of your revised paper without tracked changes. You should upload this as a separate file labeled 'Manuscript'.

We look forward to receiving your revised manuscript.

Kind regards,

Silvia Fiorelli

Academic Editor

PLOS ONE

Reviewers' comments:

Reviewer's Responses to Questions

**Comments to the Author**

1. If the authors have adequately addressed your comments raised in a previous round of review and you feel that this manuscript is now acceptable for publication, you may indicate that here to bypass the “Comments to the Author” section, enter your conflict of interest statement in the “Confidential to Editor” section, and submit your "Accept" recommendation.

Reviewer #1: All comments have been addressed

Reviewer #2: (No Response)

Reviewer #3: (No Response)

2. Is the manuscript technically sound, and do the data support the conclusions?

Reviewer #1: Yes

Reviewer #2: No

Reviewer #3: Yes

3. Has the statistical analysis been performed appropriately and rigorously? 

Reviewer #1: Yes

Reviewer #2: No

Reviewer #3: Yes

4. Have the authors made all data underlying the findings in their manuscript fully available?

Reviewer #1: Yes

Reviewer #2: Yes

Reviewer #3: Yes

5. Is the manuscript presented in an intelligible fashion and written in standard English?

Reviewer #1: Yes

Reviewer #2: No

Reviewer #3: Yes

6. Review Comments to the Author

Reviewer #1: (No Response)

Reviewer #2: This is a poorly designed study and I am not convinced that the data shown supports the claims stated in the manuscript. The responses to reviewers are vague and either miss the point or are just lightly addressed. The figures are very rudimentary and no QC plots are shown. Analyses performed are very basic and further extensive analysis are required, for instance pathway analysis. Introduction is smaller than abstract and the reasoning behind study is not comprehensively described, hence, the rationale behind the study is not clear to me. Additionally, a diagram with the workflow and inclusion/exclusion, study design/timeline would be helpful.

My main concerns are as follow:

1) the authors conclude that preoperative BMI, preoperative expression level of miR-550a-3P in peripheral blood, preoperative history of chronic pain, and postoperative NRS score are risk factors for CPSP. However, do they mean all these factors need to be present? again, the authors do not clarify that correlation does not mean causation.

2) following the questions above, The authors suggest that miR-550a-3P can be a target for post surgical pain. Would that by itself be able to prevent pain? To make this claim, authors need to perform statistical tests or experimental validation using in vitro or in vivo models.

3) the authors mention that preoperative history of chronic pain is one of the factors for development of surgical pain. This is not novel and corroborated by several other studies in the literature. Would this factor by itself influence development of CPSP? Would targeting mir-550a-3P be sufficient to counteract all the other risk factors?

4) Wouldn't it be enough to know preoperative BMI and preoperative history of chronic pain and postoperative NRS to make a patient at risk of developing pain? If the rationale is that patients will need to go through surgery anyway because of cancer, and mir-550a could be a target for treating/preventing pain - then the authors needs to show actual data that backs up this claim. Additionally, this is not clear from the beginning why look at microRNAs. I do not see how relevant it is to know that mir-550a in peripheral blood is a risk factor for CPSP if the other risk factors are predictive by themselves.

5) this sentence "Inhibition of miR-550a-3P may serve as a novel therapeutic target for chronic pain after lung adenocarcinoma occurrence" is not supported by evidence shown in the manuscript.

6) We also know from literature (see for instance Kehlet H, Jensen TS, Woolf CJ. Persistent postsurgical pain: risk factors and prevention. The lancet. 2006 May 13;367(9522):1618-25.) that genetic factors can play an important role. did the authors consider that and how do they address genetic factors in the context of the claims of this paper?

7) Authors do not address surgical technique, (or different surgeons?) - this can be a factor that influences development of pain as different techniques can target nerve fibers more

8) Inclusion/exclusion criteria do not mention pain immediately prior to surgery, which is a huge flaw of this study. Was this included under "4. Absence of nervous system dysfunction"? This is a very vague term and does not really clarify what factors contribute to it.

9) Authors did not look at pathway analysis associated with microRNAs, simply described previous literature in the Discussion.

Reviewer #3: Dear authors, here you receive my comment on the manuscript with the title ” Risk factors and related miRNA phenotypes of chronic pain after thoracoscopic surgery in lung adenocarcinoma patients “ and registration number PONE-D-23-13874.

The article is well written in understandable English. As displayed in the Methods the study with the registration number : Chi CTR2200057092 may be found in the ChiCTR database, although with slightly different title. Registration number, i.e., The risk factors and relative miRNA phenotypic analysis of chronic pain following lung adenocarcinoma surgery under VATS. As well as the site, which can be easily accessed: “http://www. medresman.org/. The authors present the results of a prospective study in 289 patients with focus on the possible relation between various risk factors among which are also possible genetic predisposition and the risk for the development of chronic pain after VATS in patients with lung carcinoma.

57 patients (=19.7%) were excluded by application of the exclusion criteria.

Did you analyze in any way how these excluded patients could have influenced your results?

Why were complications such as pulmonary complication, which is a very broad definition, and re-hospitalization both exclusion criteria? You could expect that persistent pain with secondary effects also could be one of the pulmonary complications or leading to a complication and resulting in re-hospitalization. Please comment?

Please check and comment or discuss the possible reference for miRNA’s: Sabina S, et al. Expression and Biological Functions of miRNAs in Chronic Pain: A Review on Human Studies. Int J Mol Sci. 2022 May 27;23(11):6016. doi: 10.3390/ijms23116016. PMID: 35682695; PMCID: PMC9181121. Could you explain why from all the possible miRNA’s that are in any way linked to chronic pain you chose mi R-146-3P and the other 2? This should be described more precisely in the methods?

This may provide further support as to why you chose these specific mi-RNAs and why certainly not other MiRNAs.

P4 regarding absence of nervous system disfunction: What is the definition? Were patients with preoperative well treated anxiety disorder, for instance with benzodiazepines also included or actually excluded?

P 6 regarding the statement about oxycodone sustained-release tablets: Besides the administration of these tablets twice a day, normally this is combined with the short acting opioids 4-6 times orally a day. Was this also normal procedure in your study?

P 6 last sentence “awake”: how was being awake scored?

Were there any differences in final malignancy regarding the pathological final diagnoses regarding type of differentiation of the tumour, radicality of surgical resection, primary, metastatic. See for instance, differences between smokers, non-smokers who have quit smoking and the incidence of adenocarcinoma of the lung in these patients. Furthermore, there appears to be differences in smokers vs non-smokers in relation to the development of pain (CPSP) and gene up- or down regulation. How did this may have influenced your results?

P10 table 2 did the patients have a history of smoking or quit smoking or were non-smokers?

Operation duration may be a surrogate for complexity of the procedure or tissue damage with p 0.09. Could this be related, actually the result of not being significant, to study power? Please discuss?

Were any other parameters that possibly display tissue damage or host response (e.g. inflammation) such as differences in CRP on days 1 or 2 among groups measured as you only illustrate white blood cell counts here? This is very scarse.

See regarding molecular differences within the adenocarcinoma of the lung, among others, e.g., Solis LM, et al. Histologic patterns and molecular characteristics of lung adenocarcinoma associated with clinical outcome. Cancer. 2012 Jun 1;118(11):2889-99. doi: 10.1002/cncr.26584. Epub 2011 Oct 21. PMID: 22020674; PMCID: PMC3369269. There seems to be a difference and also in outcome? Did you take this histologic difference into account?

And also among others regarding smokers and pain e.g. Oh TK, et al. Relationship between pain outcomes and smoking history following video-assisted thoracic surgery for lobectomy: a retrospective study. J Pain Res. 2018 Apr 6;11:667-673. doi: 10.2147/JPR.S157957. PMID: 29670393; PMCID: PMC5896682. How was this aspect present in your population?

See also Yoon S, Hong W, Joo H, et al. Long-term incidence of chronic postsurgical pain after thoracic surgery for lung cancer: a 10-year single-center retrospective study. Regional Anesthesia & Pain Medicine 2020;45:331-336. Here of a total of 3200 patients included in the analysis, 459 (14.3%) and 558 (17.4%) patients were diagnosed with CPSP within 3 and 36 months after surgery, respectively.

Were there any differences in duration of the ESP-block between patients or patients with failed blockade in relation to the development of pain during day 1-3 and the need for rescue medication. Please compare when possible lowest quartile vs highest quartile in the use of opioids or other analgesics?

P 7 regarding the description to hemolyze the blood cells. Please describe procedure more in detail briefly? Could it be that blood was aspirated 5-10 times fast through a very small needle to induce hemolysis in combination with the anticoagulant?

Extraction of total RNA: here both brief description of the process and reference is needed?

You mention the resistin in adipose tissue in relation to pain development. See, e.g. Majchrzak M, et al. Increased Pain Sensitivity in Obese Patients After Lung Cancer Surgery. Front Pharmacol. 2019 Jun 14;10:626. doi: 10.3389/fphar.2019.00626. PMID: 31258474; PMCID: PMC6586739. What can be said by the possibility of genetic polymorphism altering sensitivity to pain and how it may have played an important role in CPSP development in the patients of your study regarding chronic pain might related both to obesity?

and differences in gene expression? Please comment and discuss? See: Hozumi J., et al. Resistin Is a Novel Marker for Postoperative Pain Intensity. Anesthesia & Analgesia 128(3):p 563-568, March 2019. | DOI: 10.1213/ANE.0000000000003363

Textual

Provide legend for figures and explain what should be seen for our readers and what is visible and focus on the most striking features and differences between groups in the figures. Furthermore discuss these observations within the discussion in combination with chosen important references?

7. PLOS authors have the option to publish the peer review history of their article (what does this mean?). If published, this will include your full peer review and any attached files.

Reviewer #1: No

Reviewer #2: No

Reviewer #3: **Yes: **p.bruins

---

## [Author Response · Author response to Decision Letter 1]

17 Nov 2023

General response: We thank Editor and Reviewers for both the time and the insightful comments. We have carefully taken into consideration all the comments, and revised the manuscript substantially by adding analysis and clarifying content to make it more scientific and innovative. We have addressed all questions in a point-by-point manner, as shown below.

Reviewers’ comments

Reviewer#1 (No Comment)

Reviewer#2 

General comments: The figures are very rudimentary and no QC plots are shown. Analyses performed are very basic and further extensive analysis are required, for instance pathway analysis. Introduction is smaller than abstract and the reasoning behind study is not comprehensively described, hence, the rationale behind the study is not clear to me. Additionally, a diagram with the workflow and inclusion/exclusion, study design/timeline would be helpful.

Response: 

We are very grateful to the Reviewer for the constructive suggestions. We have added ROC curve for the CPSP model, and found that the prediction probability for CPSP after VATS yield the area under the receiver operating characteristic curve of 0.781 (95% CI 0.718–0.844) (Figure 5.). Thanks to Reviewer 's reminder, we have added a diagram with workflow (Figure 3.).

We have revised the "Introduction" part and listed our research purpose at the end: The primary aim of this study was to identify independently predictors of CPSP after VATS. The second aim was to search for miRNA predictors of CPSP in peripheral blood, so as to provide directions for further research on the pathogenesis and development of CPSP.

We are very sorry for the lack of pathway analysis in our clinical study. In the study, we only found several clinical risk factors for the development of CPSP and one peripheral blood microRNA indicator, and the specific mechanism needs further study in the future.

Main comments:

Comment No.1: the authors conclude that preoperative BMI, preoperative expression level of miR-550a-3P in peripheral blood, preoperative history of chronic pain, and postoperative NRS score are risk factors for CPSP. However, do they mean all these factors need to be present? again, the authors do not clarify that correlation does not mean causation.

Response: 

In this study, regression analysis was used to find several risk factors for CPSP, and collinearity diagnosis showed that there was no collinearity among these risk factors. They are independent for predicting the occurrence of CPSP, and do not need to present simultaneously. 

We are very sorry that we failed to understand the point made by the reviewer in the first comments that correlation does not mean causation. Is it because we have not conducted further extensive analysis of miR-550a-3P to determine the causal relationship between miR-550a-3P and CPSP, but we presented some inappropriate expressions in the summary at the end of the "Discussion"? With this in mind, we have replaced " Inhibition of miR-550a-3P may serve as a novel therapeutic target for chronic pain after lung adenocarcinoma occurrence." with " The expression difference of miR-550a-3P may be one of the reasons for the genetic susceptibility to CPSP in patients with thoracoscopic lobectomy, which needs to be confirmed by further studies in the future.".

Comment No.2: following the questions above, The authors suggest that miR-550a-3P can be a target for post surgical pain. Would that by itself be able to prevent pain? To make this claim, authors need to perform statistical tests or experimental validation using in vitro or in vivo models.

Response: Given the complexity of post-surgical pain, many genes might contribute. MiR-550a-3P may only be one of the related factors. We regret that no corresponding experimental validation has been carried out.

Comment No.3: the authors mention that preoperative history of chronic pain is one of the factors for development of surgical pain. This is not novel and corroborated by several other studies in the literature. Would this factor by itself influence development of CPSP? Would targeting mir-550a-3P be sufficient to counteract all the other risk factors?

Response: 

Logistic regression analysis showed that preoperative history of chronic is the risk factor for the occurrence of CPSP, and collinearity diagnosis on all the covariables included in the regression analysis showed there was no collinearity. We believe that the preoperative history of chronic pain may influence the development of CPSP by itself.

Given the complexity of post-surgical pain, many genes might contribute. In this study, we controlled some perioperative factors which may affect the occurrence of CPSP to search for CPSP susceptible population and find more closely related and more predictive genetic factors. So, expression difference of miR-550a-3P obtained in the study can well reflect the influence of genetic susceptibility on the occurrence of CPSP, but targeting mir-550a-3P alone would not be sufficient to counteract all the risk factors. (E.g. preoperative erector spinae plane block was performed to reduce the degree of acute postoperative pain. The difference of white blood cells before and after the operation did not have statistical significance could exclude the effect of postoperative infection on CPSP. All patients with early lung adenocarcinoma as their pathological diagnosis underwent video-assisted thoracoscopic lobectomy without lymph node dissection by the same group of surgeons. page 17)

Comment No.4: Wouldn't it be enough to know preoperative BMI and preoperative history of chronic pain and postoperative NRS to make a patient at risk of developing pain? If the rationale is that patients will need to go through surgery anyway because of cancer, and mir-550a could be a target for treating/preventing pain - then the authors needs to show actual data that backs up this claim. Additionally, this is not clear from the beginning why look at microRNAs. I do not see how relevant it is to know that mir-550a in peripheral blood is a risk factor for CPSP if the other risk factors are predictive by themselves.

Response: We are sorry that we did not clearly state the purpose of our research in "Introduction" at the beginning. And we have listed our research purpose at the end: The primary aim of this study was to identify independently predictors of CPSP after VATS. The second aim was to search for miRNA predictors of CPSP in peripheral blood, so as to provide directions for further research on the pathogenesis and development of CPSP.

Comment No.5: this sentence "Inhibition of miR-550a-3P may serve as a novel therapeutic target for chronic pain after lung adenocarcinoma occurrence" is not supported by evidence shown in the manuscript.

Response: We are grateful to the Reviewer for the reminder. We have replaced this sentence with " The expression difference of miR-550a-3P may be one of the reasons for the genetic susceptibility to CPSP in patients with thoracoscopic lobectomy, which needs to be confirmed by further studies in the future.".

Comment No.6: We also know from literature (see for instance Kehlet H, Jensen TS, Woolf CJ. Persistent postsurgical pain: risk factors and prevention. The lancet. 2006 May 13;367(9522):1618-25.) that genetic factors can play an important role. did the authors consider that and how do they address genetic factors in the context of the claims of this paper?

Response: Thanks for the paper provided by Reviewer. We read the review carefully, the article gives a brief description of the genetic factors. So far, several genes have been found to be associated with chronic pain, but the specific mechanism is still unclear.

Comment No.7: Authors do not address surgical technique, (or different surgeons?) - this can be a factor that influences development of pain as different techniques can target nerve fibers more.

Response: Thanks to the Reviewer for the reminder. We've added a description to "Methods". (Page 4, line 61…underwent thoracoscopic lobotomy without lymph node dissection by the same group of surgeons…)

Comment No.8: Inclusion/exclusion criteria do not mention pain immediately prior to surgery, which is a huge flaw of this study. Was this included under "4. Absence of nervous system dysfunction"? This is a very vague term and does not really clarify what factors contribute to it.

Response: We very much appreciate the Reviewer’s valuable comments and suggestions. We have added "No peripheral (somatic) or internal (visceral) chest pain before surgery" to the inclusion criteria, and replaced "Absence of nervous system dysfunction" with "Absence of peripheral neuropathy". (Page 4, lines 67,68)

Comment No.9: Authors did not look at pathway analysis associated with microRNAs, simply described previous literature in the Discussion.

Response: We very much appreciate the Reviewer’s comments. We have revised the discussion. Again, we are sorry about the lack of pathway analysis associated with microRNAs.

Reviewer#3

Comment No.1: Did you analyze in any way how these excluded patients could have influenced your results?

Why were complications such as pulmonary complication, which is a very broad definition, and re-hospitalization both exclusion criteria? You could expect that persistent pain with secondary effects also could be one of the pulmonary complications or leading to a complication and resulting in re-hospitalization. Please comment?

Response: We are very grateful to the Reviewers for their recognition of the manuscript and their constructive comments. The exclusion criteria have been reviewed and revised. (Page 4, lines 69-74) And the reasons for the selection of exclusion criteria were explained in the discussion. (Page 17, lines 295-301)

Comment No.2: Please check and comment or discuss the possible reference for miRNA’s: Sabina S, et al. Expression and Biological Functions of miRNAs in Chronic Pain: A Review on Human Studies. Int J Mol Sci. 2022 May 27;23(11):6016. doi: 10.3390/ijms23116016. PMID: 35682695; PMCID: PMC9181121. Could you explain why from all the possible miRNA’s that are in any way linked to chronic pain you chose miR-146-3P and the other 2? This should be described more precisely in the methods?

This may provide further support as to why you chose these specific mi-RNAs and why certainly not other MiRNAs.

Response: Thanks for the reviewer's suggestions. We have reviewed the reference for miRNA's: Sabina S, et al. and described the reasons for selecting mi-RNAs more precisely in the methods. (Page 9, lines 167-173)

Comment No.3: P4 regarding absence of nervous system disfunction: What is the definition? Were patients with preoperative well treated anxiety disorder, for instance with benzodiazepines also included or actually excluded?

Response: We are very sorry for our negligence, the expression of "absence of nervous system disfunction" in the inclusion criteria was wrong, we have changed it to "Absence of peripheral neuropathy". (Page 4, line 67)

Comment No.4: P6 regarding the statement about oxycodone sustained-release tablets: Besides the administration of these tablets twice a day, normally this is combined with the short acting opioids 4-6 times orally a day. Was this also normal procedure in your study?

P 6 last sentence “awake”: how was being awake scored?

Response: Thanks for Reviewer’s questions. In the study, patients were connected to intravenous analgesia pumps containing sufentanil after surgery, and oxycodone sustained-release tablets were given when patients still felt severe pain. No short acting opioids were added again. (Page 6, lines 101-104) Thanks to the reviewer's reminder, we have added the score of "awake" (with full Steward score). (Page 6, line 104)

Comment No.5: Were there any differences in final malignancy regarding the pathological final diagnoses regarding type of differentiation of the tumour, radicality of surgical resection, primary, metastatic. See for instance, differences between smokers, non-smokers who have quit smoking and the incidence of adenocarcinoma of the lung in these patients. Furthermore, there appears to be differences in smokers vs non-smokers in relation to the development of pain (CPSP) and gene up- or down regulation. How did this may have influenced your results?

P10 table 2 did the patients have a history of smoking or quit smoking or were non-smokers?

Response: Thanks for the reviewer's reminder and questions. Yes, of all the observed patients, 5 patients had different results in the final pathological diagnosis and were excluded. For clarity, we have added a separate item to the exclusion criteria for it. (Page 4, lines 70,71)

Indeed, it has been reported in the literatures that smoking status may be associated with CPSP. We re-included smoking history in the univariate analysis, but found no statistical difference between the CPSP group and non-CPSP group (P=0.212). Therefore, we no longer included smoking history in the regression analysis. (Table 2)

Comment No.6: Operation duration may be a surrogate for complexity of the procedure or tissue damage with p 0.09. Could this be related, actually the result of not being significant, to study power? Please discuss?

Were any other parameters that possibly display tissue damage or host response (e.g. inflammation) such as differences in CRP on days 1 or 2 among groups measured as you only illustrate white blood cell counts here? This is very scarse.

Response: We are very grateful to the Reviewer for the valuable comments. Considering that operation duration is likely to be related to CPSP, we adjusted the P-value limit (changed to P＜0.10) and included operation duration in regression analysis. Analysis results showed that operation duration was not a risk factor for CPSP(P=0.221). (Pages 14-16, lines 261-264, Table 6)

As mentioned by Reviewers, CRP could display inflammation. And many related studies have also conducted comparative analysis on it. But unfortunately, we missed this one in our study, which is the deficiency.

Comment No.7: See regarding molecular differences within the adenocarcinoma of the lung, among others, e.g., Solis LM, et al. Histologic patterns and molecular characteristics of lung adenocarcinoma associated with clinical outcome. Cancer. 2012 Jun 1;118(11):2889-99. doi: 10.1002/cncr.26584. Epub 2011 Oct 21. PMID: 22020674; PMCID: PMC3369269. There seems to be a difference and also in outcome? Did you take this histologic difference into account?

Response: Thanks for the literature recommended by reviewers, we have read it carefully. Whether histologic patterns and molecular characteristics of lung adenocarcinoma are related to the occurrence of CPSP is worthy of further study and analysis. At present, according to exclusion criteria 3, 4, and 5, we have excluded patients with poor prognosis or who require special treatment.

Comment No.8: And also among others regarding smokers and pain e.g. Oh TK, et al. Relationship between pain outcomes and smoking history following video-assisted thoracic surgery for lobectomy: a retrospective study. J Pain Res. 2018 Apr 6;11:667-673. doi: 10.2147/JPR.S157957. PMID: 29670393; PMCID: PMC5896682. How was this aspect present in your population?

See also Yoon S, Hong W, Joo H, et al. Long-term incidence of chronic postsurgical pain after thoracic surgery for lung cancer: a 10-year single-center retrospective study. Regional Anesthesia & Pain Medicine 2020;45:331-336. Here of a total of 3200 patients included in the analysis, 459 (14.3%) and 558 (17.4%) patients were diagnosed with CPSP within 3 and 36 months after surgery, respectively.

Were there any differences in duration of the ESP-block between patients or patients with failed blockade in relation to the development of pain during day 1-3 and the need for rescue medication. Please compare when possible lowest quartile vs highest quartile in the use of opioids or other analgesics?

Response: Thanks for the literatures recommended by reviewers. The first study showed that smoking history was not associated with postoperative pain scores, but was associated with morphine equivalent analgesics (mg) on postoperative days of 0-2. Consider that smoking history may be related to CPSP, we re-included smoking history in the univariate analysis, but found no statistical difference between the CPSP group and non-CPSP group (P=0.212). Therefore, we no longer included smoking history in the regression analysis. (Table 2)

Thanks for the reviewer's reminder and questions. The effect of nerve block may affect the development of postoperative pain. Therefore, we would test the exact effect of the ESP-block after completion. (Page 5, lines 92-95) And, failure to block would be excluded. (Exclusion criteria, page 4, line 74)

Thanks for the reviewer's suggestions. We compared the dosage of sufentanil, remifentanil and dexmedetomidine per kilogram during anesthesia. The results showed no statistical difference between CPSP group and non-CPSP group. (Table 2) Rescue medication was associated with postoperative NRS score. However, no correlation was allowed between the covariables included in the regression analysis, so we did not make additional comparison in rescue medication.

Comment No.9: P7 regarding the description to hemolyze the blood cells. Please describe procedure more in detail briefly? Could it be that blood was aspirated 5-10 times fast through a very small needle to induce hemolysis in combination with the anticoagulant?

Extraction of total RNA: here both brief description of the process and reference is needed?

Response: We very much appreciate the Reviewer’s reminder and have added the corresponding description. (Page 7, lines 137,138)

We are sorry that we did not quite understand the comments about the "Extraction of total RNA". We have described the whole process of Extraction of total RNA in the manuscript. (Pages 7-8, lines 144-157) And total RNA was extracted from white cells using mirVanaTM RNA Isolation Kit according to the manufacturer’s speciﬁcations. So, no more references were provided.

Comment No.10: You mention the resistin in adipose tissue in relation to pain development. See, e.g. Majchrzak M, et al. Increased Pain Sensitivity in Obese Patients After Lung Cancer Surgery. Front Pharmacol. 2019 Jun 14;10:626. doi: 10.3389/fphar.2019.00626. PMID: 31258474; PMCID: PMC6586739. What can be said by the possibility of genetic polymorphism altering sensitivity to pain and how it may have played an important role in CPSP development in the patients of your study regarding chronic pain might related both to obesity?

and differences in gene expression? Please comment and discuss? See: Hozumi J., et al. Resistin Is a Novel Marker for Postoperative Pain Intensity. Anesthesia & Analgesia 128(3):p 563-568, March 2019. | DOI: 10.1213/ANE.0000000000003363

Response: We very much appreciate the Reviewer’s valuable comments. And we have re-written this part according to the Reviewer’s suggestion. (page18, lines 319-326)

Comment No.11: Textual

Provide legend for figures and explain what should be seen for our readers and what is visible and focus on the most striking features and differences between groups in the figures. Furthermore, discuss these observations within the discussion in combination with chosen important references?

Response: Thanks for the reviewer's suggestion. We have presented two main resulting diagrams in the manuscript (Figure 4 and Figure 5), and explained the striking features and differences between groups in the figures. (page15, lines 266-274) And, we have revised the discussion according to the Reviewer’s suggestion.

---

## [Decision Letter · Decision Letter 2]

12 Jan 2024

Risk factors and related miRNA phenotypes of chronic pain after thoracoscopic surgery in lung adenocarcinoma patients

PONE-D-23-13874R2

Dear Dr. He,

We’re pleased to inform you that your manuscript has been judged scientifically suitable for publication and will be formally accepted for publication once it meets all outstanding technical requirements.

Kind regards,

Silvia Fiorelli

Academic Editor

PLOS ONE

Additional Editor Comments (optional):

Congratulations to the authors and thanks to the reviewers for the provided suggestions which really helped improve the quality of the manuscript

Reviewers' comments:

Reviewer's Responses to Questions

**Comments to the Author**

1. If the authors have adequately addressed your comments raised in a previous round of review and you feel that this manuscript is now acceptable for publication, you may indicate that here to bypass the “Comments to the Author” section, enter your conflict of interest statement in the “Confidential to Editor” section, and submit your "Accept" recommendation.

Reviewer #3: All comments have been addressed

Reviewer #4: All comments have been addressed

2. Is the manuscript technically sound, and do the data support the conclusions?

Reviewer #3: Yes

Reviewer #4: Yes

3. Has the statistical analysis been performed appropriately and rigorously? 

Reviewer #3: Yes

Reviewer #4: Yes

4. Have the authors made all data underlying the findings in their manuscript fully available?

Reviewer #3: Yes

Reviewer #4: Yes

5. Is the manuscript presented in an intelligible fashion and written in standard English?

Reviewer #3: Yes

Reviewer #4: Yes

6. Review Comments to the Author

Reviewer #3: Dear authors,

here you receive my comment on the manuscript with the title ” Risk factors and related miRNA phenotypes of chronic pain after thoracoscopic surgery in lung adenocarcinoma patients “ and registration number PONE-D-23-13874R2.

The article is well written in understandable English. You have adequately addressed all my previous comments. I have no further comments.

Reviewer #4: The authors have responded to prior comments in an appropriate, informative manner, we appreciate that. The added diagrams/figures have enhanced the manuscript.

-

7. PLOS authors have the option to publish the peer review history of their article (what does this mean?). If published, this will include your full peer review and any attached files.

Reviewer #3: No

Reviewer #4: No

---

## [Editor Report · Acceptance letter]

3 Mar 2024

PONE-D-23-13874R2 

PLOS ONE

Dear Dr. He, 

I'm pleased to inform you that your manuscript has been deemed suitable for publication in PLOS ONE. Congratulations! Your manuscript is now being handed over to our production team.

Kind regards, 

on behalf of

Dr. Silvia Fiorelli 

Academic Editor

PLOS ONE